# Does the Level of Training Interfere with the Sustainability of Static and Dynamic Strength in Paralympic Powerlifting Athletes?

**Felipe J. Aidar** [1,2,3,4], **Stefania Cataldi** [5], **Georgian Badicu** [6], **Ana Filipa Silva** [7,8],
**Filipe Manuel Clemente** [7,8,9], **Valerio Bonavolontà** [5], **Gianpiero Greco** [5,*], **Márcio Getirana-Mota** [1,2]
**and Francesco Fischetti** [5]

1   Graduate Program of Physical Education, Federal University of Sergipe (UFS),
    São Cristovão 49100-000, Brazil; fjaidar@gmail.com (F.J.A.); marcio_getirana@hotmail.com (M.G.-M.)
2   Group of Studies and Research of Performance, Sport, Health and Paralympic Sports (GEPEPS),
    Federal University of Sergipe (UFS), São Cristovão 49100-000, Brazil
3   Department of Physical Education, Federal University of Sergipe (UFS), São Cristovão 49100-000, Brazil
4   Graduate Program of Physiological Science, Federal University of Sergipe (UFS),
    São Cristovão 49100-000, Brazil
5   Department of Basic Medical Sciences, Neuroscience and Sense Organs, University of Study of Bari,
    70124 Bari, Italy; stefania.cataldi@uniba.it (S.C.); valerio.bonavolonta@uniba.it (V.B.);
    francesco.fischetti@uniba.it (F.F.)
6   Department of Physical Education and Special Motricity, Faculty of Physical Education and Mountain Sports,
    Transilvania University of Braşov, 500068 Braşov, Romania; georgian.badicu@unitbv.ro
7   Escola Superior Desporto e Lazer, Instituto Politécnico de Viana do Castelo, Rua Escola Industrial e Comercial
    de Nun'Álvares, 4900-347 Viana do Castelo, Portugal; anafilsilva@gmail.com (A.F.S.);
    filipe.clemente5@gmail.com (F.M.C.)
8   Research Center in Sports Performance, Recreation, Innovation and Technology (SPRINT),
    4960-320 Melgaço, Portugal
9   Instituto de Telecomunicações, Delegação da Covilhã, 1049-001 Lisboa, Portugal
*   Correspondence: gianpierogreco.phd@yahoo.com

**Abstract:** Background: Paralympic powerlifting (PP) presents adaptations that the training tends to provide, mainly concerning the mechanical variables. Objective: Our aim was to analyze mechanical, dynamic and static indicators, at different intensities, on the performance of paralympic powerlifting athletes. Methods: 23 athletes of PP, 11 national level (NL) and 12 regional level (RL) performed dynamic and static tests over a comprehensive range of loads. The study evaluated regional and national level athletes and the influence on the training level on the performance of strength. The study was carried out in four weeks, with the first week to familiarize with the one repetition maximum (1RM), day 1, and there was a 72-h rest and familiarization with dynamic and static tests carried out day 2. In week 2, the 1RM tests were performed (day 1 and 72 h later), and the static tests were performed with a distance of 15 cm from the bar to the chest, with the tests of maximum isometric strength, time to maximum isometric strength, rate of force development (RFD), impulse, variability and fatigue index (IF) taking place on day 2. In weeks three and four dynamic tests were performed, including means propulsive velocity, maximum velocity, power and prediction of one maximum repeat. Results: Differences were found, with better results than for RL in relation to NL in MVP (45%, 55%, and 75% 1RM), in VMax (50%, 55%, 75% and 95% 1RM). In power, the NL had better results (40%, 45%, 50%, 60% and 95% 1RM). Conclusion: RL athletes tend to present better results with regard to velocity, however in power, NL athletes tend to present better performances.

**Keywords:** muscle strength; force-velocity; disabled persons; athletic performance; paralympic powerlifting

## 1. Introduction

One of the questions that has been asked in strength training is related to the quantification and monitoring of the load, aiming at better performance. The most used variables in this sense have been type and order of exercise, intensity or load, number of repetitions, and series and rest between sets [1]. The manipulation of these variables has usually been used as a training control [2,3]. Thus, the training load has been determined from the relative load (% of the maximum load of one repetition, 1RM), being the main factor of control and determination of the intensity and fatigue relative to the strength training [4,5]. Although these variables are used to control training, their use can induce excess fatigue and mechanical and metabolic tension [6–10].

Thus, the evaluation of an athlete's training status and initial condition is the crucial point for the correct elaboration of a training program to be applied in different phases of the sports preparation [11]. In these initial conditions, some training variables are manipulated to prescribe and control resistance training programs such as: sets, time intervals, position and intensity [11–13]. On the other hand, in these initial conditions, there are possible disparities between different training methods to determine mechanical outputs in strength-power exercises. Therefore, the velocity-based approach to training becomes a practical and effective alternative for coaches [11,14,15]. This statement is supported by current studies that emphasize that training control based on the percentage of one repetition maximum would have low control and could still lead to sub or super dimensional planning. In this direction, training control through speed would be more indicated [8,11,14,15].

In this direction, the bench-press (BP) is one of the most studied exercises on measures of power, strength, and speed [14]. BP has been shown to be closely correlated with sporting success as a multi-joint exercise that mimics various sporting actions [11,15]. More specifically, in paralympic powerlifting (PP), where the bench press is the only exercise used, being an adaptation of the conventional powerlifting bench press [16]. It is noteworthy that in the PP the athletes have their lower limbs extended over the bench, in view of several eligible disabilities [16], although the differences between the PP and conventional powerlifting are still not clear [14]. In addition, the study in relation to this modality has been growing and the use of mechanical variables has been used [13,14,17].

On the other hand, paralympic powerlifting (PP) has particularities notably in relation to elite athletes in relation to the possible adaptations that training tends to provide, especially in relation to mechanical variables [14,18]. As a strength modality, PP training involves several variables, such as strength (Ability to oppose resistance), power (product of force and velocity), volume, intensity, bar displacement speed (distance divided by time), among others [14,19]. Due to this specificity, it is suggested that powerlifting athletes undertake specialized training programs to obtain non-specific adaptations in relation to strength and velocity of movement [20].

In PP there is only one functional classification category, and all eligible disabilities are physical and compete together, with division only into body weight categories [16]. However, research with PP athletes has evaluated the origin of injuries and functional classification principles of athletes [21–23]. In addition, studies investigating relevant aspects regarding the force-velocity of these athletes and aspects that tend to influence performance are not yet clear in the literature [21–23]. Therefore, the aim of this study was to analyze mechanical, dynamic and static indicators, at different intensities, on the performance in regional and national level athletes of paralympic powerlifting. From the above, we raised the following study hypotheses: stronger athletes would generate more speed with the same load, and strength-velocity-based assessment could be a way of controlling and evaluating performance in paralympic powerlifting athletes.

## 2. Materials and Methods

### 2.1. Experimental Approach to the Problem

The study was carried out in four weeks. The first week was intended to familiarize with the tests of one maximum repetition (1RM) and with dynamic and static tests. In week

two, 1RM tests and static tests were performed, including maximum isometric force (MIF), time to MIF (Time), rate of force development (RFD), impulse, variability and fatigue index (FI). At weeks three and four, dynamic tests were performed, including mean propulsive velocity (MPV), maximum velocity (VMax), power (Power) and prediction of one repetition maximum (PredRM).

Figure 1 exemplifies the experimental design of the study.

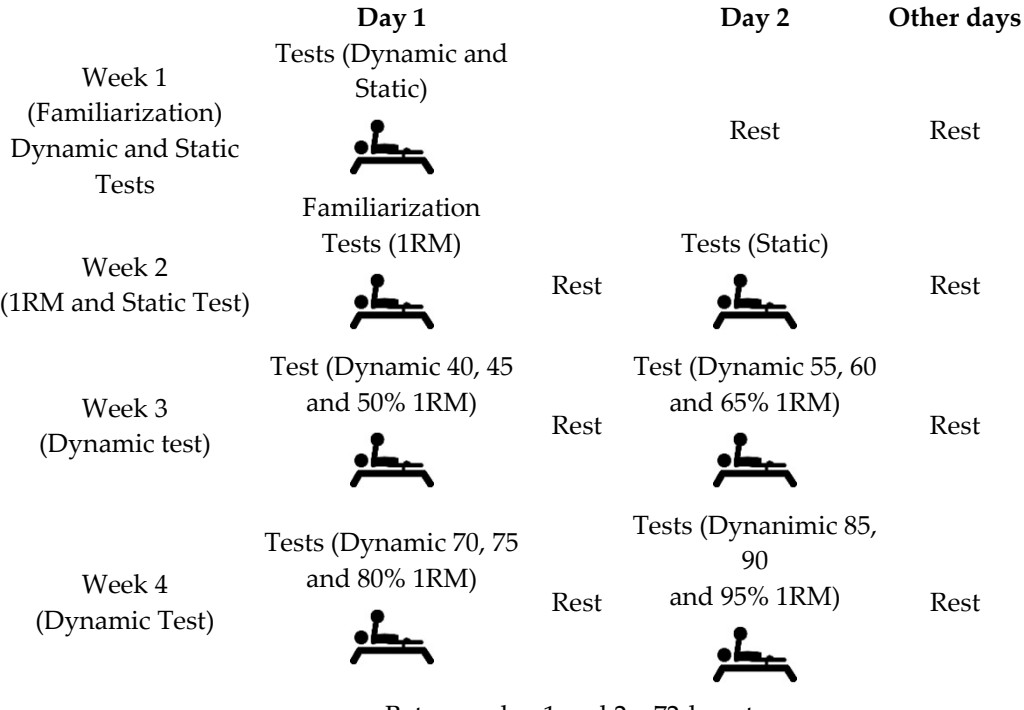

**Figure 1.** Study design.

### 2.2. Sample

The sample consisted of 23 male Paralympic powerlifting athletes (PP), 11 at the national level (NL) and 12 at the regional level (RL). All of them were competitors and were part of the extension project at the Federal University of Sergipe, Sergipe, Brazil. All were eligible to compete in the sport [16] and NL athletes are ranked among the top ten in their respective categories and with a minimum of 24 months of experience in the sport. In the NL group, four athletes with spinal cord injury below the eighth thoracic vertebra; two with polio, one with cerebral palsy, and four amputees. In the RL group, a training experience in the modality of a maximum of 12 months was attributed [19]. Five subjects with spinal cord injury due to accidents with injuries below the eighth thoracic vertebra; three with amputation, two with polio, and two with atrogriposis. The sample characterization is shown in Table 1.

**Table 1.** Sample characterization.

|  | Regional Level | National Level | *p* | ICC | CV | α |
|---|---|---|---|---|---|---|
| Age (years) | 29.25 ± 4.50 | 26.13 ± 7.22 | 0.223 | 0.307 | 3.07 | 0.001 |
| Body weight (kg) | 80.86 ± 15.36 | 82.80 ± 31.73 | 0.852 | 0.537 | 0.17 | 0.644 |
| Experience (years) | 0.4 ± 0.18 | 4.45 ± 0.31 | 0.001 # | 0.014 | 12.12 | 0.506 |
| 1RM Bench Press Test (kg) | 87.43 ± 17.24 | 136.88 ± 28.65 | 0.001 # | 0.021 | 11.17 | 0.168 |
| 1RM/Body mass | 1.08 ± 0.36 | 1.65 ± 0.32 * | 0.007 # | 0.037 | 7.78 | 0.271 |

# $p < 0.05$ (independent "t" test). * Values above 1.4 in the bench press would be considered elite athletes, according to Ball and Wedman [24].

The sampling power was calculated a priori using the open-source software G*Power® (Version 3.0; Berlin, Germany), choosing a "F family statistics (ANOVA)" con-sidering a standard $\alpha < 0.05$, $\beta = 0.80$ and the effect size of 1.33 found for the Rate of Force Development (RFD) in Paralympic powerlifting athletes in the study by Sampaio et al., [12]. Thus, it was possible to estimate a sample power of 0.80 (F (2.0): 4.73) for a minimum sample of eight subjects per group, suggesting that the sample size of the present study has statistical strength to respond to the research approach.

The athletes participated in the study on a voluntary basis and signed a free and informed consent form, in accordance with resolution 466/2012 of the National Research Ethics Commission (CONEP), of the National Health Council, in accordance with the ethical principles expressed in Helsinki Declaration (1964, reformulated in 2013), by the World Medical Association. This study was approved by the Research Ethics Committee of the Federal University of Sergipe, CAAE: 2.637.882 (date of approval: 7 May 2018).

### 2.3. Instruments

The determination of body mass was performed using a Michetti Wheelchair Weighing Scale (Michetti, São Paulo, SP, Brazil) to facilitate the weighing of them seated, with a maximum supported weight capacity of 300 kg and a dimension of $1.50 \times 1.50$ m. In the evaluation, an official adapted bench press (Eleiko Sport AB, Halmstad, Sweden) was used, according to the norms of the International Paralympic Committee (IPC, 2020). The bar was made by Eleiko, 220 cm (Eleiko Sport AB, Halmstad, Sweden), weighting 20 kg [13,16].

### 2.4. Procedures

#### 2.4.1. Load Determination

Data were collected at a time outside of important competitions, and the athletes' training was related to their experience (RL and NL), as mentioned above. Athletes completed a baseline measurement session to assess 1RM in the bench press using an official bench and IPC Olympic bar (Eleiko Sport AB, Halmstad, Sweden) approved by the International Paralympic Committee [16]. The 1RM test was conducted, and each subject started the trials with a weight they believed that they could lift once, using maximum effort. Weight increments were then added until they reached the maximum load that could be lifted once. If the participant could not perform a single repetition, 2.4% to 2.5% was subtracted from the load used in the test. The subjects rested for 3 to 5 min between attempts [25,26].

The test was preceded by a warm-up set (10 to 12 repetitions) with approximately 50% of the load to be used for the first attempt of the 1RM test. The testing started two minutes after the warm-up. The load recorded as 1RM was the one when the individual could complete only one repetition. The form and the adapted technique used in the performance of each attempt was standardized and continuously monitored to ensure the quality of the data. The test for determining 1RM was performed at week one.

#### 2.4.2. Warm-Up

The warm-up for upper limbs, using three exercises (abduction of the shoulders with dumbbells, elbow extension in the pulley and rotation of the shoulders with dumbbells) with three sets of 10 to 20 repetitions [27,28]. Soon after, a specific warm-up was performed on the bench press with a 30% load of 1RM, 10 slow repetitions (3:1 s, eccentric: concentric) and 10 fast repetitions (1:1 s, eccentric: concentric). Followed with five sets of bench press of five maximum repetitions (5 sets—85 at 90% RM), using a fixed load. During the test, athletes received verbal encouragement in order to achieve maximum performance [27,28]. To perform the bench press, an official straight bench (Eleiko Sport AB, Halmstad, Sweden), approved by the International Paralympic Committee [16] was used.

### 2.4.3. Dynamic Evaluation

The athletes were evaluated during the competitive phase of the season and were familiar with the testing procedures due to their constant training and testing routines. Athletes were instructed to perform the movement as fast as possible. An official paralympic powerlifting bench (Eleiko Sport AB, Halmstad, Sweden) was used during the measurements. The 1RM bench press test was performed on a paralympic powerlifting bench (Eleiko Sport AB, Halmstad, Sweden), following standard procedures reported in other studies [18]. To measure the velocity of movement, a valid and reliable linear position transducer Speed4Lift (Speed4Lift®, Madrid, Spain) [29] was attached to the bar [18,28]. The highest averages of bars, mean of propulsion velocity (average values only of the propulsive phase, positive acceleration, that is, above the acceleration of gravity) and peak velocity (peak distance/time ratio), Power (force × velocity) and Prediction of 1 Repetition Maximum (MPV, VMax, Power, PredRM, respectively) were used for analysis purposes. The predicted 1RM was determined by the Bench Press equation provided in the Speed4lift device (Speed4Lift®, Madrid, Spain) [29].

### 2.4.4. Isometric Force Measurements

The static variables of force were rate of force development (RFD), maximum isometric force (MIF) (N), fatigue index (FI) (%) and time to MIF (time) (m/s), were determined by a Chronojump force sensor (Chronojump, BoscoSystem, Barcelona, Spain) [17], with a capacity of 500 kg, output impedance of $350 \pm 3$ ohm, insulation resistance greater than 2000 cc, input impedance $365 \pm 5$ ohm, analog converter 24-bit 80 Hz digital. The equipment was attached to the bench press, using Spider HMS Simond carabiners (Simmond, Chamonix, France), with a load of 21 kN, (Union Internationaledes Associations d'Alpinisme-UIAA). A steel chain with a load of 2300 kg was also used, used to fix the force sensor to the bench press. The distance from the force sensor to the center of the joint was used to determine torques and other values [18,28]. Maximum isometric strength (MIF) was determined by the maximum strength of the upper limbs, and an elbow angle close to 90 ° was maintained, and at a distance of 15 cm from the bar to the chest. Athletes were instructed to make a single maximum movement (as fast as possible). The fatigue index (FI) was determined in the same way as the MIF, where the athletes maintained the maximum contraction for 5.0 s. The FI was calculated by the formula: FI = ((final MIF − initial MIF/final MIF) × 100). The RFD was calculated by the force/time ratio (RFD = Δ force/Δ time) [18,28]. The instruments used in the evaluations are shown in Figure 2.

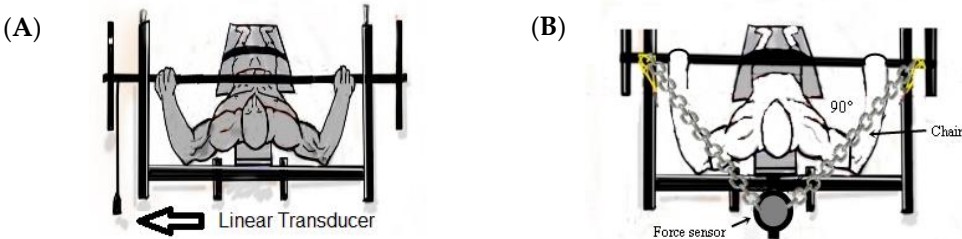

**Figure 2.** Instruments used in the evaluations: (**A**) dynamic (linear transducer) and (**B**) static (force sensor).

### 2.5. Statistics

Descriptive statistics were performed using measures of central tendency, mean (X) ± Standard Deviation (SD) and 95% confidence interval (95% CI). To verify the normality of the variables, the Shapiro Wilk test was used, considering the sample size. Data for all variables analyzed were homogeneous and normally distributed. To evaluate the strength indicators of the groups and percentage of 1RM, the ANOVA (Two Way) test was performed with Bonferroni's Post Hoc. Pearson's "r" was used for correlation, with the following cut-off points: 0.00–0.10, Negligible correlation; 0.10–0.39, Weak correlation; 0.40–0.69, Moderate correlation; 0.70–0.89, Strong correlation and 0.90–1.00, Very strong

correlation [30]. To check the effect size, (partial Eta squared: η2p), adopting values of low effect (≤0.05), medium effect (0.05 to 0.25), high effect (0.25 to 0.50) and very high effect (>0.50) for ANOVA [31,32]. For the t test, an effect size (Cohen's d) was considered, adopting values of low effect (≤0.20), medium effect (0.20 to 0.80), high effect (0.80 to 1.20) and very high effect (>1.20) [33,34]. Statistical analysis was performed using the computerized Statistical Package for the Social Science (SPSS), version 22.0 (IBM, North Castle, New York, NY, USA). The significance level adopted was $p < 0.05$.

## 3. Results

The results found for the average propulsive velocity (m/s) in the regional and national levels, in the percentages from 40 to 65% and 70 to 95% of 1RM, are found in Figure 3.

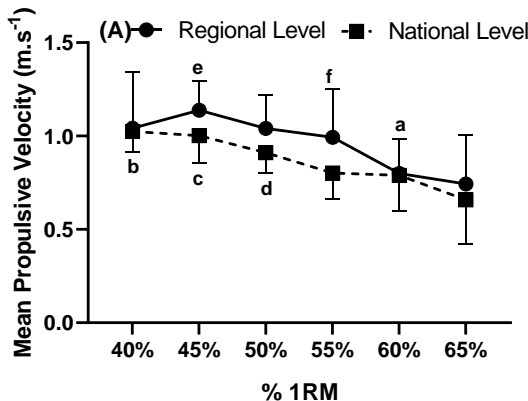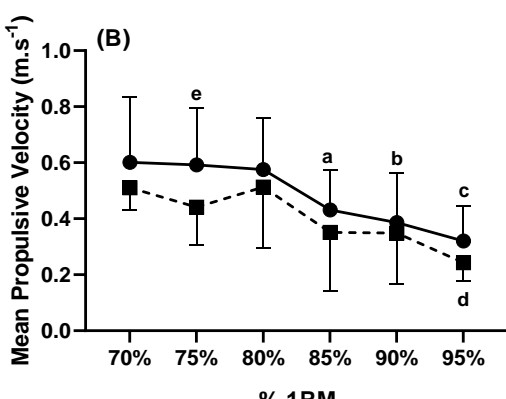

**Figure 3.** Mean propulsive velocity (m·s$^{-1}$) measured from (**A**) mean propulsive velocity (m·s$^{-1}$) measured from 40 to 65% of 1RM and (**B**) mean propulsive velocity (m·s$^{-1}$) measured from 70 to 95% of 1RM in national and regional levels. (**A**): a: Indicates difference in RL between 60% compared to 45% ($p = 0.043$) and 50% 1RM ($p = 0.041$); b: Indicates differences in NL between 40% compared to 55% ($p = 0.001$), 60% ($p = 0.005$) and 65% 1RM ($p = 0.002$); c: Indicates differences in NL between 45% compared to 55%, 60% and 65% 1RM ($p < 0.001$); d: Indicates differences in NL between 50% compared to 55% ($p = 0.030$) and 65% 1RM ($p = 0.016$); e: Indicates differences in percentage 45% between RL and NL ($p = 0.041$); f: Indicates differences in percentage 55% between RL and NL ($p = 0.047$). The value of F = 19.119, and η2p = 0.705 was very high effect (IntraClass) and F = 1.104, and η2p = 0.121 medium effect (InterClass). (**B**): a: Indicates difference in RL between 85% compared to 75% 1RM ($p = 0.039$); b: Indicates differences in RL between 90% compared to 75% ($p = 0.031$), 80% 1RM ($p = 0.023$); c: Indicates differences in RL between 95% versus 70% ($p = 0.005$), 75% ($p = 0.005$), 80% 1RM ($p = 0.004$), and 85% 1RM ($p = 0.023$); d: Indicates differences in NL between 95% versus 70% ($p < 0.001$), 75% ($p = 0.004$) and 80% 1RM ($p = 0.036$); e: Indicates differences in the percentage 75% between RL and NL ($p = 0.020$). The value of F = 25.224, and η2p = 0.759 was very high effect (IntraClass) and F = 1.606, and η2p = 0.167 medium effect (InterClass).

The results found in the maximum velocity (m·s$^{-1}$) in the regional (RL) and national levels (NL), in the percentages of 40 to 65% and 70 to 95% of 1RM, are found in Figure 4.

The results found in the power (W) in the regional level (RL) and national level (NL), in the percentages from 40 to 65% and 70 to 95% of 1RM, are found in Figure 5.

The results found in the maximum predicted repetition (kg) in the regional level (RL) and national level (NL), in the percentages from 40 to 65% and 70 to 95% of 1RM, are shown in Figure 6.

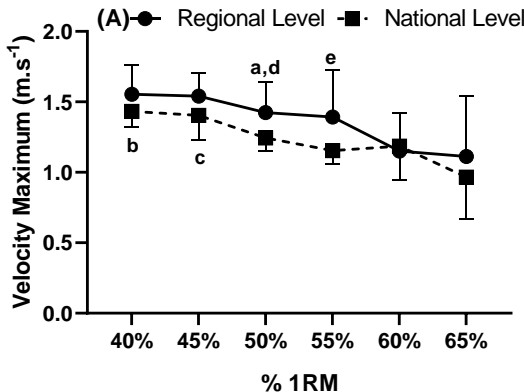
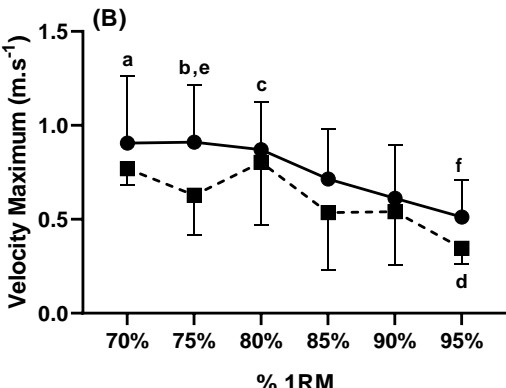

**Figure 4.** Maximum velocity (m·s$^{-1}$) measured from (**A**) maximum velocity (m·s$^{-1}$) measured from 40 to 65% of 1RM and (**B**) maximum velocity (m·s$^{-1}$) measured from 70 to 95% of 1RM in national and regional levels. (**A**): a: Indicates difference in RL between 60% compared to 50% 1RM ($p = 0.030$); b: Indicates differences in NL between 40% versus 50% ($p = 0.020$), 55% ($p < 0.001$) and 65% 1RM ($p = 0.008$); c: Indicates differences in NL between 45% compared to 55 and 65% ($p < 0.001$), and 60% 1RM ($p = 0.007$); d: Indicates differences in percentage 50% between RL and NL ($p = 0.041$); e: Indicates differences in percentage 55% between RL and NL ($p = 0.049$). The value of F = 20.390, and η2p = 0.718 was very high effect (IntraClass) and F = 1.087, and η2p = 0.120 medium effect (InterClass). (**B**): a: Indicates difference in RL between 70% compared to 95% 1RM ($p = 0.008$); b: Indicates differences in RL between 75% compared to 85% ($p = 0029$) and 95% 1RM ($p = 0.016$); c: Indicates differences in RL between 80% compared to 90% ($p = 0.019$) and 95% 1RM ($p = 0.001$); d: Indicates differences in NL between 95% versus 70% ($p < 0.001$), 75% ($p = 0.006$) and 80% 1RM ($p = 0.041$); e: Indicates differences in percentage 75% between RL and NL ($p = 0.013$); f: Indicates differences in the percentage 95% between RL and NL ($p = 0.040$). The value of F = 17.916, and η2p = 0.691 very high effect (IntraClass), and F = 1.302, and η2p = 0.140 medium effect (InterClass).

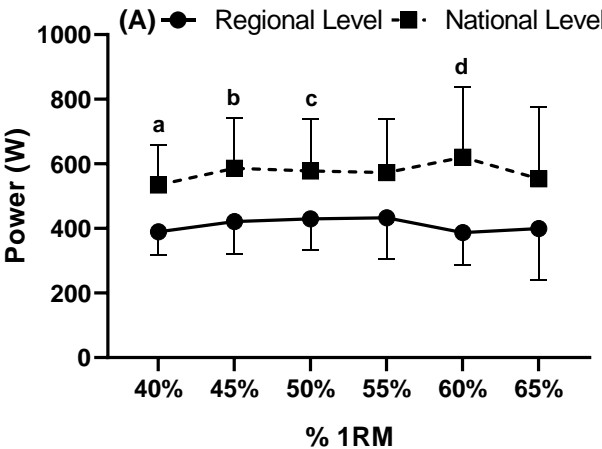
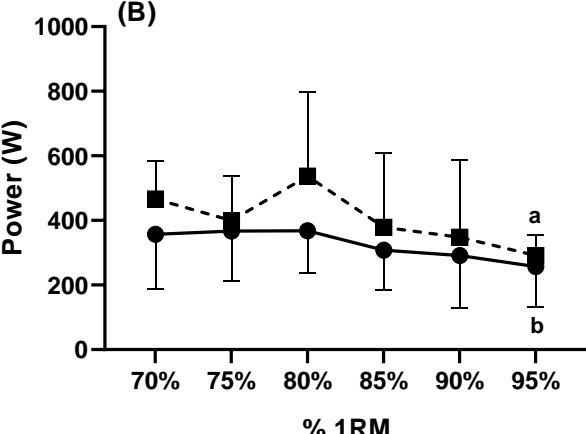

**Figure 5.** Power (W) measured from (**A**) power (W) measured from 40 to 65% of 1Re (**B**) power (W) measured from 70 to 95% of 1RM in national and regional level. (**A**): a: Indicates differences in percentage 40% between RL and NL ($p = 0.004$); b: Indicates differences in percentage 45% between RL and NL ($p = 0.004$); c: Indicates differences in percentage 50% between RL and NL ($p = 0.023$); d: Indicates differences in percentage 60% between RL and NL ($p = 0.032$). The value of F = 7.254, and η2p = 0.476 was high effect (InterClass). (**B**): a: Indicates differences in RL between 95% compared to 70% ($p = 0.038$), 75% ($p = 0.049$) and 80% 1RM ($p = 0.004$); b: Indicates differences in NL between 95% compared to 70% 1RM ($p = 0.007$). The value of F = 7.218, and η2p = 0.474 was high effect (IntraClass).

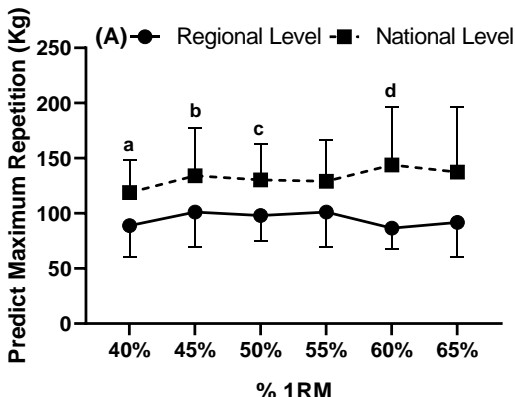
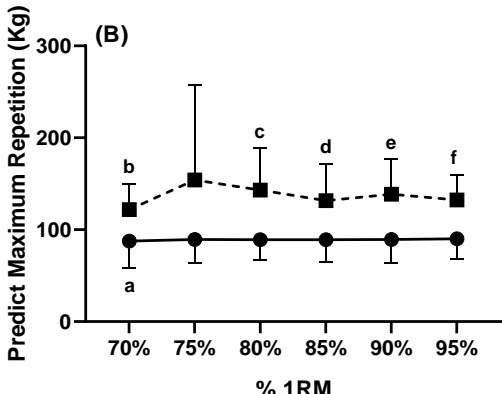

**Figure 6.** Maximum predicted repetition (kg) measured from (**A**) maximum predicted repetition (kg) measured from 40 to 65% of 1RM and (**B**) maximum predicted repetition (kg) measured from 70 to 95% of 1RM in national and regional level. (**A**): a: Indicates differences in percentage 40% between RL and NL ($p = 0.009$); b: Indicates differences in percentage 45% between RL and NL ($p = 0.016$); c: Indicates differences in percentage 50% between RL and NL ($p = 0013$); d: Indicates differences in percentage 60% between RL and NL ($p = 0.021$). The value of F = 8.653, and η2p = 0.520 was very high effect (InterClass). (**B**): a: Indicates differences in NL between 95% compared to 70% 1RM ($p = 0.007$); b: Indicates differences in the percentage 70% between RL and NL ($p = 0.046$); c: Indicates differences in the percentage 80% between RL and NL ($p = 0.016$); d: Indicates differences in percentage 85% between RL and NL ($p = 0.021$); e: Indicates differences in the percentage 90% between RL and NL ($p = 0.013$); f: Indicates differences in the percentage 95% between RL and NL ($p = 0.007$). The value of F = 10.741, and η2p = 0.573 was very high effect (IntraClass) and F = 0.594, and η2p = 0.069 medium effect (InterClass).

The results found in dynamic dynamic mechanical variables (VMP, V max, pot and 1RM) and isometric (MIF, rime, RFD, impulse, variability, IF) of the regional and national level athletes are shown in Table 2.

**Table 2.** Dynamic and isometric strength indicators (mean ± standard deviation, 95% CI) in regional level and national levels.

| | Regional Level | National Level | *t* | *p* | Cohen's *d* |
|---|---|---|---|---|---|
| MPV (m·s$^{-1}$) | 0.21 ± 0.10 | 0.13 ± 0.06 | 1.984 | 0.065 | 0.97 [c] |
| VMax (m·s$^{-1}$) | 0.38 ± 0.14 | 0.27 ± 0.09 | 2.057 | 0.056 | 0.93 [c] |
| Power (W) | 170.78 ± 89.70 | 182.22 ± 106.93 | 0.246 | 0.003 * | 1.69 [d] |
| 1RM (kg) | 86.78 ± 19.86 | 129.22 ± 29.56 | 3.576 | 0.809 | 0.12 [a] |
| MIF (N) | 658.94 ± 185.27 | 971.64 ± 122.59 | 4.223 | 0.001 * | 1.99 [d] |
| Time (μs) | 2167.95 ± 1280.83 | 2894.60 ± 1368.17 | 1.163 | 0.262 | 0.55 [b] |
| RFD (N·s$^{-1}$) | 2208.68 ± 885.42 | 2826.28 ± 1384.03 | 1.128 | 0.276 | 0.53 [b] |
| Impulse (N·s) | 2922.76 ± 838.02 | 4440.89 ± 652.23 | 4.289 | 0.001 * | 2.02 [d] |
| Variability (N) | 51.74 ± 21.54 | 32.84 ± 15.84 | 2.120 | 0.049 * | 1.00 [c] |
| FI (%) | 12.32 ± 5.65 | 9.16 ± 3.71 | 1.412 | 0.180 | 0.66 [b] |

\* $p < 0.05$ (ANOVA two-way, and Bonferroni post hoc test). a: small effect, b: medium effect, c: high effect, d: very high effect. **Legend:** MPV: mean propulsive velocity, VMax: maximum velocity, Power: Power, 1RM, one repetition maximum, MIF: maximum isometric force, Time: time to MIF, RFD: rate of force development, FI: fatigue index. **Note:** Dynamic variables were evaluated with 100% of 1 Rpetition Maximum.

Table 3 shows the correlations between predicted maximum repetition (PredRM) and static (MIF), in relation to one repetition maximum (1RM).

**Table 3.** Correlation between predicted values in different percentages in relation to the absolute load (1RM) in regional and national level athletes in the paralympic bench press (mean ± standard deviation).

| Load | Regional Level (kg) | Pearson's $r$ | National Level (kg) | Pearson's $r$ |
|---|---|---|---|---|
| 40% Pred 1RM | 88.89 ± 28.70 | 0.451 * | 118.78 ± 29.92 | 0.810 # |
| 50% Pred 1RM | 98.00 ± 23.01 | 0.439 * | 130.33 ± 32.99 | 0.893 # |
| 60% Pred 1RM | 86.56 ± 18.53 | 0.576 ** | 143.89 ± 52.61 | 0.696 ** |
| 70% Pred 1RM | 87.56 ± 28.88 | 0.710 # | 121.78 ± 27.91 | 0.963 ## |
| 80% Pred 1RM | 89.22 ± 22.03 | 0.949 ## | 143.00 ± 45.74 | 0.744 # |
| 90% Pred 1RM | 89.33 ± 25.25 | 0.920 ## | 138.56 ± 38.49 | 0.824 # |
| MIF (kgF) | 67.26 ± 18.88 | 0.575 ** | 98.92 ± 12.50 | 0.585 ** |

* Weak correlation, ** Moderate correlation, # high correlation and ## Very high correlation. 1RM in Regional Level: 86.78 ± 19.86 and National Level: 129.22 ± 29.56. Legends: Pred 1RM: repetition maximum predict, 1RM: one repetition maximum.

## 4. Discussion

This study was designed to analyze mechanical, dynamic and static indicators of strength at different intensities on the performance of regional and national level athletes in paralympic powerlifting. The results found reveal that regional level athletes impose higher velocity in all loads. In particular, some intensities are found differences for the mean propulsive velocity 45%, 55% and 75% of 1RM and when analyzed the maximum velocity in the percentages 50%, 55%, 75% and 95% of 1RM, in comparison with national level athletes. Statically when evaluated MIF and RFD, there is a difference in MIF from national to regional level. When analyzing the RFD, there were no differences between the national and regional levels. However, when power is evaluated, the national level developed higher power rates than regional level in loads of the 40%, 45%, 50%, 60% and 95% of 1RM. In the results on maximum velocity and average propulsive velocity as well as in the maximum isometric force, the regional level athletes presented lower performance values than the national level athletes [35]. In this direction, when training with very high loads, the longer the time required to overcome an absolute load, the greater the effort performed [35], situation similar to that found in our study, where with higher loads, the velocity tends to decrease. Regarding the VMP and VMax, the regional level athletes tended to present greater effort for the same loads, when compared with the national level athletes, in view of the higher velocity presented. In contrast, other studies have found no differences in velocity between national level athletes compared to regional level athletes [36–38].

A point to be analyzed and discussed was to evaluate why athletes who have absolutely higher strength values cannot print higher velocity than athletes who have lower strength. Regardless of whether they are regional or national level athletes, NL athletes performed better in specific percentages of the 1RM (VMP: 45%, 55%, and 75%) (VMax: 50%, 55%, 75%, and 95%), leading to a perspective of adaptation in relation to the National Level loads. In this direction, the average propulsive velocity can be used as a performance marker and has been shown to be more reliable than static indicators [20]. On the other hand, static indicators of force, such as RFD, would be an effective form of control [39], since the generation of force in a short period of time would be of great importance in the maximum force generation, and through an encoder this variable does not tend to be evaluated correctly.

With regard to RFD, Zemková, Poór and Pecho [40] identified an interesting finding. The authors identified that individuals with higher RFD tend to obtain higher performance for power with lower loads, while individuals with higher FIM tend to produce higher power, however, using higher loads. Although the study was not performed on the bench press, nor with Paralympic athletes, these findings are contrary to our findings. In our study, trained athletes had higher FIM than Regional Level athletes, but did not show higher RFD. The differences were present in the NL in relation to the RL in the loads between 40% and 60% of 1RM. For the others, the differences were not significant. Demonstrating that NL

athletes, despite having more strength, did not have a higher RFD, that is, they take longer to develop strength than RL athletes.

When analyzing the performance on power production among athletes, Aidar et al. [13], when comparing conditions of execution of the adapted bench press, a significant difference was found favorable for the NL athletes in relation to the RL for the load of 40% of 1RM Corroborating our finding, Miller et al. [41], also found results similar to our findings, corroborating our study. The authors identified that the maximum power produced between trained and untrained men was 40% versus 60% respectively. These findings tend to indicate that training with higher loads tends not to be the most suitable for power development. This information brought by the authors, in Figure 4, where our athletes lost performance with loads closer to 1RM. These findings emphasize that lower loads would be more suitable for power development [41].

These differences in power between trained and untrained athletes can be explained by the training time and the specificity of strength training, since athletes train with very high loads and do not have power as a factor linked to performance. In this study, it was reported that strength training would provide a decrease in the threshold of muscle fiber recruitment and provide an increase in the rate of discharge during submaximal contractions for the same motor units. The authors suggest that muscle strength gains can be attributed to an increase in excitatory synaptic input or to adaptations in motor neuron properties. Thus, athletes with longer training time would present a higher discharge rate during contractions, which would provide a greater advantage in power production than athletes with shorter training time [42], which was not observed in our study.

On the other hand, a review indicated that the regional level would not need to emphasize specific power training but rather strength training, and that experienced athletes can emphasize power development while maintaining their strength levels [43]. This manifestation of force would be influenced by several aspects. In this sense, the main variables that would affect the power would be the force-velocity relationship and the length-tension relationship. These variables would be directly manipulated by morphological factors, which would directly affect the individual's ability to generate force quickly [44]. Thus, these factors would be related to the fiber type of the muscle area, architectural characteristics of the muscle and properties of the tendon, as well as neural factors, including motor unit recruitment, firing frequency, synchronization and intermuscular coordination [45,46].

The aforementioned changes tend to be promoted by strength training. However, to increase power, it is not enough to increase maximum strength [47,48]. Due to the specifics of generating maximum force for some sports, in the shortest possible time (milliseconds or even ≤300 ms). This time would not allow maximum force to be reached [38]. Therefore, the development of power, being a relationship between movement velocity and force, and the two, where these two variables tend to present a linear relationship [49–51], that is, if the velocity were higher in the generation of power, the force would tend to be smaller and vice versa. This would explain the fact that our NL athletes have greater force and lower velocity and the RL have the opposite, which is a possible adaptation to training with high loads.

Among all of the specificities presented about power development, long-term development would be linked to the integration of various strength training techniques [52–54]. This would probably be a justification for why more experienced athletes would develop higher levels of power than more inexperienced athletes. This would occur due to the adaptations that are promoted by the strength training itself, as well as by the probable form of training that is used.

Regarding fatigue, there were no differences between national and regional athletes ($p = 0.180$), these findings corroborate another study that evaluated the manifestation of strength in different types of disabilities, where there were no differences in fatigue between the different types of disability. In our study, in absolute terms, athletes at the National level showed less fatigue when compared to athletes at the regional level [18]. Our findings regarding fatigue, indicated a low fatigue (between 9% and 12%) according to the training

level. Fatigue has been the subject of many studies for strength gain. Corroborating this, one study demonstrated that there were no differences between high and low fatigue training in terms of isometric strength. Thus, fatigue does not seem to be a critical stimulus for strength gain [7].

Furthermore, our study has some limitations, despite the relevance of the results found. The sample consisted of national and regional athletes with different disabilities eligible for the modality. In this sense, the findings are for Paralympic powerlifting practitioners, not looking for the specifics of different physical disabilities eligible for the modality. However, the findings are relevant to coaches and researchers for a greater understanding of strength training and the relationship of strength to velocity and other strength indicators in Paralympic powerlifting athletes, and their effects on sport performance. Another limitation raised is linked to the fact that the evaluation was performed acutely, that is, in a single training session, so the results could be different when evaluating weeks or even longer periods of training.

## 5. Conclusions

Paralympic powerlifting (PP) is characterized by training with high loads, greater than 80% of 1RM. Our findings indicate that barbell velocity in PP was higher in RL athletes compared to NL athletes. Thus, due to the characteristics of the sport, the specific training, adapted to the rules of the sport, tends to provide a more effective control of the bar, which ends up promoting a lower speed of execution in athletes of national level. On the other hand, with regard to power and predicted 1RM, it was higher in NL athletes when compared to RL. In this sense, national level athletes presenting reduced velocity, this indicates that strength, in these athletes, would have a greater importance in the generation of power, mainly in higher loads, demonstrating the specific adaptation to force provided by maximum strength training.

In relation to the results found, and given the fact that stronger athletes tend to generate more strength against the same resistance, I could advise coaches that training with lower loads, with an emphasis on movement velocity, could provide improvements in athletes' strength, even national athletes. Thus, strength-velocity-based assessment appears to be the sustainable method of monitoring and evaluating performance in athletes, including paralympic powerlifting athletes.

On the other hand, other studies should evaluate other deficiencies and their impact on the velocity and strength of paralympic athletes, since the bases of balance and movement execution in the adapted bench press tend to be different in each type of physical disability.

**Author Contributions:** Conceptualization, F.J.A. and F.M.C.; methodology, G.G. and G.B.; software, F.M.C.; validation, F.J.A., S.C. and F.F.; formal analysis, A.F.S.; investigation, S.C. and V.B.; resources, F.J.A. and M.G.-M.; data curation, F.M.C. and G.G.; writing—original draft preparation, F.J.A., S.C. and G.B.; writing—review and editing, F.M.C., G.G. and F.F.; visualization, V.B.; supervision, F.M.C., G.G. and F.F.; project administration, A.F.S. All authors have read and agreed to the published version of the manuscript.

**Funding:** This research received no external funding.

**Informed Consent Statement:** Informed consent was obtained from all subjects involved in the study.

**Data Availability Statement:** The data that support this study can be obtained from the address: www.ufs.br/Department of Physical Education, accessed on 7 January 2022.

**Conflicts of Interest:** The authors declare no conflict of interest.

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
