# Peer review of "Does the Level of Training Interfere with the Sustainability of Static and Dynamic Strength in Paralympic Powerlifting Athletes?"

_sustainability, doi:10.3390/su14095049_

Round 1

Reviewer 1 Report

The manuscript entitled « Does the level of training interfere with the sustainability of static and dynamic strength in Paralympic Powerlifting athletes? » aimed to analyze mechanical, dynamic and static indicators, at different intensities, on the performance in different tests (Maximum Isometric Strength, Time to Maximum isometric strength, Rate of Force Development, Impulse, Variability and Fatigue Index) of Paralympic Powerlifting athletes.

The manuscript is quite well written, with adequate references, but the presentation, form, typos and English must be copyedited (same remark concerning the keywords, Table 3, “Figura”, references => commas…).

I suggest to improve the following points:

Transitions between the paragraphs in the introduction part of the manuscript are needed for better clarity and for the readers.

The author must evoke the hypotheses at the end of the introduction.

The eventuality of a session effect should possibly be discussed and/or evoke as a limitation of this study.

The results could be discussed with other researches (e.g., Folland et al., 2002; Teles et al., 2021; Wilcox, 2019).

The authors must present all the results concerning the ANOVAs, t test.

The use of letter is very hard to read and understand for the readers, please envisage other presentation of the results in the Figures

Recommendations:

The manuscript must be carefully check (I suggest to use expert proofreaders)

Certain parts of the manuscript in the abstract, introduction, materials and methods, discussion and conclusion are highlighted in yellow, please remove…

Abstract:

“Differences were found, with better results than for RL in relation to NL in MVP (45, 55, 60, 75, 85, 90 and 95% 1RM), in VMax (50, 55, 70, 75 and 80% 1RM) » please check for the English

The authors must check for the typos example lines 15, 16 …

Keywords:

“Muscle strength” is written twice, please removed one.

Introduction:

The sentence “Thus, the intensity of has been used as the most important stimulus related to strength training [4] and has been used from the relative load (% of the maximum load of a repetition, 1RM) [5]” must be rewrote.

Transitions are needed between the different parts of the introduction.

The authors must clearly indicate the different hypotheses according to the test performances and athlete’s levels.

Method:

The sex of the participant must be added in the “sample” part of the manuscript

Please replace FIM by MIF page 5.

Figure 1:

Please put (1 RM) in the same line than (Dynamic and Static)

Please add “rest” between “Day 1” and “Day 2” for better clarity

Data analysis section:

Analysis:

Did the authors use a pre-experiment power analysis? What was the sample size recommended? If it haven’t been done, please evoke this point as a limitation (=> sample size)

Results:

The use of letter is very hard to read and understand for the readers, please envisage other presentation of the results in the Figures 2 to 5

All the results concerning the ANOVAs, t test must be presented (values of F and t).

Please replace “Figura 2” by “figure” same remark concerning Figura 4 and Figura 5.

Table 3: please check for the typo

Figure 4 : B) there is no difference between Regional Level and National Level ? even for 80% please clarify.

Discussion

Please check for the English: first paragraph.

The results evoked in the first paragraph (i.e., “average propulsive velocity 45, 55 and 75% of 1RM and when analyzed the maximum velocity in the percentages 50, 55, 75 and 95% of 1RM) must be discussed according to the litterature in the same paragraph. Please include the second paragraph in the first one for better comprehension.

Results concerning “Fatigue” must be discussed (e.g., Folland et al., 2002; Wilcox, 2019)

I suggest to discuss the results according to recent research work:

Teles, L.J.L.; Aidar, F.J.; Matos, D.G.d.; Marçal, A.C.; Almeida-Neto, P.F.d.; Neves, E.B.; Moreira, O.C.; Ribeiro Neto, F.; Garrido, N.D.; Vilaça-Alves, J.; Díaz-de-Durana, A.L.; Clemente, F.M.; Jeffreys, I.; Cabral, B.G.d.A.T.; Reis, V.M. Static and Dynamic Strength Indicators in Paralympic Power-Lifters with and without Spinal Cord Injury. Int. J. Environ. Res. Public Health 202118, 5907. https://doi.org/10.3390/ijerph18115907

The absence of power differences especially between 70% and 95% of 1RM between beginners and trainers must be more deeply discussed/explained

Limitation:

I suggest to discuss the weak sample size and the possible session effect (due to repetition)

Reviewer 2 Report

To be readable, the english must be significantly improved: many errors and missing words all along the text and in the abstract.

I some point it miss citations (e.g. ...research with PP athlete has evaluated the origin of injuries and functional.....please cite some references.

No reference for the Michetti test.

Missing references for all the instruments used for force/power  assessment with validation of the systems.

Legends of figures are poorly understadable, instead of using desctiption (a =, b =, ecc) please make tables with data and with significant differences .

Where are the results for ANOVA ?

Discussion is very confused. Organize it in paragraphs for each variables considered.

Results are very poor:

Our findings indicate that the barbell velocity in PP wa higher in RL....

This is not a great finding, and it can be an effect merely of the larger loads lifted by NL athletes.

The increased power of course produce lower speed. This is not new.

The conclusion are confusing and poorly understandable, neither provide something new.

Reviewer 3 Report

Congratulations for your manuscript. I consider that it adds relevant information to the scientific literature. Nonetheless, I have some questions that you should resolve in order to improve your study.

  • The use of abbreviations throughout the document should be reviewed. From the first time the abbreviation is used, it must always be used. This does not occur in the document and makes it difficult to read and understand.
  • I reccomend to improve the resume section, you should include more precise information.
  • Paragraph 2: The concept of strength-power should be defined. Also, you should explain why is speed-based training important, and which have been the contribution? You should include some references that support this statement.
  • Paragraph 3 and 4: You should improve both paragraphs. I consider that is important to clarify the concept of power, velocity, and force-velocity profile. In addition, can you explain what do you mean force-velocity profile? Also, are there differences in the strength variables in the bench press between Paralympic and conventional powerlifting? If so, please indicate them.
  • Paragraph 5: More information should be provided on the functioning of Paralympic powerlifting and the different types of injuries of the participants, with their possible influence on performance depending on the type and degree. Indicate if there are categories and how they are organized.
  • Figure 1: You should include the reference through the text instead of an isolate paragraph.
  • In the description of the group NL repeats “two with polio”. According to the pathologies of the participants, 12 athletes are described, although in the sample 11 are indicated for the NL group. Please, try to explain it.
  • Why is a maximum of 12 months experience an inclusion criterion in the RL group? It is not clear if athletes of different levels (NL and RL) or athletes with different experience (experts and non-experts) are compared. This should be cleared up. One thing and another are not the same. Is it possible that there are athletes with a higher level (NL) even if they have a lower degree of experience or not? Please, explain it.
  • This information in the legend of table 1 has already been exposed previously: All athletes with loads that keep them in the top 10 of their categories nationwide. It is redundant, please choose one of them.
  • Were all the athletes in a similar state of form, or they have a similar training program? Considering it, try to explain if the comparison in the strength evaluation is reliable? At what time of the season was the evaluation carried out? Some of this information is provided in the procedure section and should be placed in the description of the method. The description of the participants should be improved regarding to these points.
  • In the procedure section, the information on the material previously provided is repeated. Needs to be reviewed, information is redundant (example: Eleiko Sport AB, Halmstad, Sweden)
  • You should add a reference in relation because the participants rested 3 to 5 minutes between attempts to determinate 1RM. Please, justify it.
  • Two different warm-ups are done, one for the determination of 1RM and one for the dynamic evaluation. It should be explained and justify the reason for it?
  • You must provide the validation reference of the Speed4Lift device.
  • Can you include a photograph or graph with the execution of the dynamic evaluation and another of the static evaluation where the position of the devices is observed?
  • In the evaluation of the dynamic force, the procedure is not specified correctly. Can you provide what kind of information was given to the participants regarding the performance, velocity, or execution' movement? Also, were participants asked to perform all repetitions as fast as possible?
  • The legend of the table should be organized just below the table and in a single paragraph, so that it will be easier to interpret with the figure, explaining what the subscript letters mean. The results obtained in the analysis must be incorporated into the body of the text of the results section.
  • Figure 2 and figure 3: Can you explain why the Mean Propulsive Velocity and maximum velocity with 80% load is greater than a load with 75%? Is this an error, or is it logical that this is so?
  • Please, check the units of the variables through vertical axis of each figure.
  • In table 2, could you provide the values of the dynamic force variables regarding % 1RM was obtained? Also, you should indicate it in the legend of the table to improve the description.
  • You should analise that the participants in NL group had greater power values in
  • The second paragraph of the discussion should be revised. I recommend improving accuracy and clarifying in relation to the relationship between perform the movement faster or slower but with higher loads and their implication for intensity and variables such as power.
  • I strongly recommend revising and rewrite the discussion section for a better understanding. Also, you should focus on the results obtained through this section.

Round 2

Reviewer 2 Report

pag. 14, line 7 to 12 are in portuguese language. Is beautiful, but please translate in english.

Author Response

Reviewer: pag. 14, line 7 to 12 are in portuguese language. Is beautiful, but please translate in english.

Reply: Thank you for your suggestion. We have highlighted the changes in green.